# Acceptability of COVID-19 Vaccination among Greek Health Professionals

**DOI:** 10.3390/vaccines9030200

**Published:** 2021-02-28

**Authors:** Dimitrios Papagiannis, George Rachiotis, Foteini Malli, Ioanna V. Papathanasiou, Ourania Kotsiou, Evangelos C. Fradelos, Konstantinos Giannakopoulos, Konstantinos I. Gourgoulianis

**Affiliations:** 1Public Health & Vaccines Laboratory, Faculty of Nursing, School of Health Sciences, University of Thessaly, 41500 Larissa, Greece; 2Department of Hygiene and Epidemiology, Faculty of Medicine, School of Health Sciences, University of Thessaly, 41222 Larissa, Greece; grach@uth.gr; 3Respiratory Disorders Laboratory, Faculty of Nursing, University of Thessaly, 41500 Larissa, Greece; fmalli@uth.gr; 4Community Nursing Lab, Faculty of Nursing, University of Thessaly, 41500 Larissa, Greece; iopapathanasiou@uth.gr; 5Faculty of Nursing, University of Thessaly, 41500 Larissa, Greece; okotsiou@uth.gr (O.K.); efradelos@uth.gr (E.C.F.); 6Medical Association of Larissa, “Hippocrates”, 41222 Larissa, Greece; artgiann@gmail.com; 7Respiratory Medicine Department, Faculty of Medicine, University of Thessaly, 41500 Larissa, Greece; kgourg@med.uth.gr

**Keywords:** Covid-19, vaccination, acceptability, health care workers

## Abstract

Health Care Workers are at the front line of the fight against Covid-19. The aim of this study was to evaluate the acceptability of vaccination against COVID-19 among health professionals (physicians, dentists, pharmacists) two weeks prior to the start of the Greek vaccination campaign against COVID-19. A cross-sectional online survey was conducted over the period 15–22 December 2020 in 340 health professionals in Central Greece. We found a high level of acceptance for COVID-19 vaccine (78.5%) and a high vaccination coverage for the influenza vaccine (74%). Age > 45 years (OR = 2.01; 95% C.I. = 2.01−4.3), absence of fear over vaccine safety (OR = 4.09; 95% C.I. = 1.36–12.3), and information received from the Greek public health authorities (OR = 11.14; 95% C.I. = 5.48–22.6), were factors independently associated with the likelihood of COVID-19 vaccination acceptance. Our study indicates a high level of the COVID-19 vaccination acceptance among physicians, dentists and pharmacists. Nevertheless, several interventions can be implemented to increase acceptance of vaccine among health-care workers (HCWs) and could be especially directed at younger and vaccine-hesitant health care workers due to fear of vaccine side-effects. Last, our results provide some evidence that receiving vaccine-related information from the Greek Center for Diseases Control (E.O.D.Y.) could reduce the drivers of hesitancy and enhance the acceptability of COVID-19 vaccination.

## 1. Introduction

Health-care workers (HCWs) are vital resources for every country. It is also well known that protecting health of HCWs is an important pillar of pandemic preparedness [1]. During the COVID 19 pandemic, HCWs are at the front line in terms of risk of infection and death. From the beginning of the COVID-19 pandemic, healthcare workers have demonstrated professional dedication despite a fear of becoming infected and infecting patients or family members [2]. Vaccination is an effective approach to prevent infection and reduce mortality of many infectious diseases [3]. Furthermore, frontline health care workers are a high priority target group for vaccination against COVID-19 [4]. Since December 2020, several vaccines have been authorized in the European Union (E.U.) [5,6]. The vaccination started in E.U members—including Greece—on 27 December 2020 and HCWs have been identified as a priority group for vaccination against COVID-19 [7]. The aim of our study was to evaluate the acceptability of COVID 19 vaccination among physicians, dentists, and pharmacists one week prior to the start of the COVID-19 vaccination campaign in Greece.

## 2. Materials and Methods

A cross-sectional online survey was conducted among the members of the Larissa, (Thessaly, Greece) Medical, Dentist and Pharmacist associations one week before the official start of national vaccination campaign against the SARS CoV 2 in Greece from 15 to 22 December 2020. Participation was voluntary. The members of the three associations were invited to participate in the online survey. The participants provided anonymous informed consent on the survey platform before they could proceed to the completion of the questionnaire. A structured questionnaire was created. Τable 1 depicts the basic socio-demographic information of the participants. Out of 340 participants, 174 were males (51.2%) and 166 females (48.8%), and the mean age of the study population was 47.7 years (SD = 10.97). The distribution according to work status of the participants was as follows: (213) of the participants (62.7%) were Physicians, 80 were Dentists (23.5%), and 47 (13.8%) were Pharmacists, Table 1. 

The questionnaire (Table 2) included questions on demographics, the value, effectiveness, and safety of vaccines. In addition, the questionnaire included questions on the willingness to accept the COVID-19 vaccination and the vaccination coverage for the seasonal influenza vaccine (flu season 2020–21). Acceptability of COVID-19 vaccination was measured with one item: Are you going to be vaccinated against COVID-19 when the vaccine will be available? (Yes/No). In the case of COVID-19 vaccination refusal the participants were requested to report the reason of non-vaccination acceptance. All participants were asked to report if they had fear over COVID-19 vaccination side effects, and if they believed that the COVID-19 vaccines have been developed in a short time. In addition, the respondents were asked to evaluate the quality of COVID-19 vaccine-related information from Greek public health authorities. Vaccination coverage against influenza was measured with the following question: Have you been vaccinated with the influenza vaccine (season 2020–2021)? (Yes/No). Moreover, the participants were asked about their sources of information on the safety of COVID-19 vaccines. The protocol of the study was approved by the Scientific Committee of the University Hospital of Larissa (Protocol number: 53988; 15/12/2020).

### 2.1. Sample Size Calculation

The sample size was calculated by the use of the following formula, provided by the Open Epi, Version 3 software.
Sample size *n* = [DEFF ‘x’ Np (1 − *p*)]/[d^2^/Z^2^1 − α/2 ‘x’(N − 1) + *p* ’x’ (1 − *p*)](1)
where *n* is the required sample size. N is Population size (for finite population correction factor or fpc) = 2100; where *p* is the hypothesized (%) frequency of outcome factor in the population (50% +/−5; Confidence limits as % of 100) (absolute +/−%)(d) = 5%; where DEFF (design effect = 1); where (Z is a constant = 1.96. for 95% C.I.). On the base of the above assumptions the minimum required sample size calculated was 325 participants.

### 2.2. Statistical Analysis

All data were tabulated using the SPSS 21.0 software. Absolute (*n*) and relative frequencies (%) were presented for qualitative variables, while quantitative variables were presented as mean (standard deviation). Chi-square test or Fischer’s exact test was used for the univariate analysis of qualitative variables and Student’s *t*-test for quantitative variables. The majority of independent variables were initially coded in a four option Likert scale (Fully agree/agree/disagree/fully disagree). These four response options were collapsed into a dichotomous (binary) variable (Fully agree/agree vs. disagree/fully disagree). Variables found to be statistically significant in the univariate analysis (*p* < 0.05) were included in a stepwise binary logistic regression analysis model, in order to explore potential independent associations with respondents’ intention to accept vaccination against COVID-19 [8]. In this model COVID-19 vaccination acceptance was the dependent variable and factors found to be significant (*p* < 0.05) in univariate analysis were the independent variables. Age was transformed to a categorical variable based on the cut-off level of mean age value. Adjusted odds ratio (OR) and 95% confidence intervals (95% CI) were calculated. The level of statistical significance was set at 0.05.

## 3. Results

Overall, 214 out of the 1500 members of the Larissa Medical Association were participated in (Response rate: 15%). The corresponding rate for Dentists and Pharmacists Association was 24% (81/300) and 15% (45/300), respectively. 

Dentists reported the highest percentage for Covid-19 vaccine acceptability (82.5%) followed by the physicians (80%) and the Pharmacists (64.5%). The majority of the participants reported that the vaccines are generally safe and effective tools for the protection of public health. We have also recorded high rate of acceptance for vaccination against Covid-19 disease (78.5%) and high vaccination coverage (74%) against influenza (influenza season 2020–2021). About the main source of information, the participants reported that the biomedical scientific publications was the primary source followed by the International Health Organizations (WHO, CDC, ECDC), the Greek CDC (Table 2).

Among HCWs who refused vaccination the main reasons reported were fear of side effects (67/73; 93%), and a belief that the development time of the vaccines against COVID-19 was short (69/72; 96%). Univariate analysis results (Table 2) show that older age (mean age for vaccination acceptance = 45.67 years vs. 41.3 for vaccination refusal), positive perceptions on the safety (80% vaccination acceptance among participants who reported that vaccines are safe vs. 5.5% acceptance among those who reported that vaccines are not safe), history of influenza vaccination during 2020–2021 season (subjects who have received a flu shot reported significantly higher COVID-19 vaccination acceptance than those who remained unvaccinated against flu), fear over side effects of COVID-19 vaccination (people who were concerned over COVID-19 vaccination side effects, recorded significantly lower COVID-19 vaccination acceptance than people with no fear over side-effects), and COVID-19 vaccine-related information obtained from the Greek public health authority (participants who considered the information provided by the Greek Public Health Authorities as reliable showed an almost 3-fold prevalence of vaccination acceptance in comparison to those who rated the provided information as non-reliable), were significantly associated with the prevalence of vaccination acceptance. Stepwise binary logistic regression analysis (Table 3) revealed that age > 45 years (OR = 2.01; 95% C.I. = 2.01−4.3), absence of fear over vaccine safety (OR = 4.09; 95% C.I. = 1.36–12.3) and information received from the Greek public health authorities (OR = 11.14; 95% C.I. = 5.48–22.6) were independently associated with the likelihood of COVID-19 vaccination acceptance. With respect to source of information about COVID-19 vaccination, our results showed that subjects who obtained information from social media/blogs demonstrated low prevalence (although insignificant) of COVID-19 vaccination acceptance.

## 4. Discussion

HCWs are considered to be the most trusted source of vaccine-related information for patients and the public. They are in the best position to understand hesitant patients, to respond to their safety concerns, and to find ways of explaining them the substantial benefits of vaccination [9]. The present study demonstrates a high vaccination acceptance rate for COVID-19 (80 %) among physicians. This figure is substantially higher in comparison to a previous study conducted in the same region during February 2020 [10].

Our results are in line with a study from Israel which reported a 78% vaccination acceptance among physicians. This is the also the case with a French study during the first wave of COVID-19, which found that 81.5% of the participated healthcare workers expressed their intention to be vaccinated when a vaccine will become available [11]. Increasing age was identified as an independent predictor of vaccination acceptance. This complies well with published evidence [11] and it is plausible given that increasing age is associated with high risk of comorbidities and could lead to a higher risk perception in comparison to health care workers at younger age groups. In addition, recent meta-analytic evidence suggests that age is an important predictor of COVID-19 vaccination acceptance [12].

Fear over vaccine side effects had a negative impact on COVID-19 vaccination acceptance. Fear over vaccination-related side effects was the main reason for non-acceptance of COVID-19 vaccination. This is in line with accumulated evidence that the major reason for hesitation or refusal was fear of side effects [12]. We have also recorded a high vaccination coverage for the influenza vaccine (74%). This figure is unprecedented for healthcare workers in Greece [13,14]. Notably, influenza vaccination (season 2020–2021) was not found to be an independent predictor of COVID-19 acceptance in multivariate analysis. This finding contradicts the results of a recently published study among HCWs, from France and French-speaking parts of Belgium and Canada which reported an independent effect of previous influenza vaccination (season 2019–2020) on the acceptability of COVID-19 vaccination (71.6%) [13].

Nevertheless, it should be mentioned that in our survey we assessed the impact of the vaccination coverage of the current influenza season (2020–2021) on vaccination acceptability [15]. We believe that the unprecedented influenza vaccination coverage may be have prevented this variable for reaching statistical significance in logistic regression analysis. The pharmacists are health care professionals playing an important role in the provision of pharmaceutical care to the patients. The low acceptance of COVID-19 vaccine by the pharmacists if confirmed by future research could be an issue of concern for policy makers and public health authorities.

The present study has several limitations, which should be taken into account prior to the interpretation of the results. First, our study has the limitation of being a cross-sectional study, and this type of study design cannot inform us about causal relationships. Second, this is a questionnaire-based study, consequently, information bias may have occurred. Further, we believe that the acceptance rate of the COVID-19 vaccine found in our study could be overestimated given that healthcare professionals who were not interested in the vaccination may not have been motivated to participate in our survey. Nevertheless, subjects who believe that vaccination against COVID-19 is an obvious choice may also have been less inclined to participate in than subjects who are concerned over vaccine safety. An additional shortcoming is related to the possible limited (regional) generalizability of our results. Last, we did not obtain information from other important health care groups (e.g., nurses).

## 5. Conclusions

In the present study we report considerably high acceptance of the COVID-19 vaccination among Greek HCWs (physicians, dentists, pharmacists). Nevertheless, several interventions can be implemented to increase acceptance of vaccine among HCWs, and could be especially directed at younger and hesitant health care workers due to fear of vaccine side-effects. Last, our results provide evidence that receiving vaccine-related information from the Greek Center for Diseases Control (E.O.D.Y.) could reduce the drivers of hesitancy and enhance the acceptability of COVID-19 vaccination.

## Figures and Tables

**Table 1 vaccines-09-00200-t001:** Demographic characteristics of the health care professionals.

	N/Total (%) or Mean ± SD
**Characteristics**	
Total sample	340
Sex	
Male	173/340 51%
Female	167/340 49%
Age (years)	44.7 + 10.97
Education	
Bachelor degree	340/340 100%
MSc	97/340 28.5%
PhD	65/340 19.1%
**Occupation**	
Physician	214/340 63%
Dentist	81/340 24%
Pharmacist	45/340 13%

**Table 2 vaccines-09-00200-t002:** Univariate analysis of Covid-19 vaccination acceptability.

Variables	N, (%), Mean/SD	Acceptance of COVID 19 Vaccine		
		Yes (%)	No (%)	*p* Value
Sex				
Male		142/173 82	31/173 18	0.053
Female		125/167 75	42/167 25	
Age		267 45.67 + 11.00	72 41.3 + 11.01	0.003
Occupation				
Physicians		172/214 80	42/214 20	0.018
Dentists		66/80 82.5	14/80 17.5	
Pharmacists		29/45 64	16/45 36	
**Q1.** Overall the vaccines are important for Public Health				
Fully agree/Agree	336/340 99	251/336 75	85/336 25	0.0005
Fully disagree/Disagree	4/340 1	1/4 25	3/4 75	
**Q2.** Overall, the vaccines are safe.				
Fully agree/Agree	331/340 97	267 /331 80	69/331 20	<0.001
Fully disagree/Disagree	9/340 3	1/9 11	8/9 89	
**Q3.** The vaccines in general are effective.				
Fully agree/Agree	328/340 96.5	264/328 80	64/328 20	<0.001
Fully disagree/Disagree	12/340 3.5	2/12 16.5	10/12 83.5	
**Q4.** I am concerned over COVID-19 vaccination side effects				
Fully agree/Agree	194/340 57	126/194 65	67/194 35	0.001
Fully disagree/Disagree	104/340 30.5	99/104 95	5/104 5	
**Q5.** The information I have received on vaccination against COVID-19 from the Greek Public Health authorities is reliable.				
Fully agree/Agree	230/340 68	211/230 92	19/23 8	0.001
Fully disagree/Disagree	81/340 24	29/81 35	52/81 65	
**Q6.** The development time of the vaccines against COVID-19 was short.				
Fully agree/ Agree	261/340 77	191/261 73	69/261 27	0.002
Fully disagree/Disagree	52/340 15	49/52 94	3/52 6	
**Q7.** Have you been vaccinated with the influenza vaccine 2020–2021?				
Yes	251/340 74	207/251 82.5	43/251 17.5	0.001
No	89/340 26	60/89 67	29/89 33	
**Q8.** What was your main source of information on COVID-19 vaccines?				
-Biomedical scientific publications				0.220
-Website of a Greek CDC		215/286 75	71/286 25	
-International Health Organizations (WHO, CDC, ECDC)				
-Television/radio/newspapers				
-Social media/blogs		35/50 70	15/50 30	

**Table 3 vaccines-09-00200-t003:** Binary logistic regression analysis of COVID-19 vaccination acceptability.

Independent Variable	OR	95% C.I.	*p* Value
Age			
>45 years	2.01	1.02–4.3	0.045
≤45 years	1.00 (ref)		
No Fear over side effects	4.09	1.36–12.3	0.012
Reliable information from Greek CDC (E.O.D.Y.)	11.14	5.48–22.6	0.000

## Data Availability

The data sets used and/or analyzed during the present study are available from the corresponding author on reasonable request.

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
