# Peer review of "Acceptability of COVID-19 Vaccination among Greek Health Professionals"

_vaccines, 2021, doi:10.3390/vaccines9030200_

Round 1

Reviewer 1 Report

This is a cross-sectional survey of 340 healthcare workers (physicians, dentists, and pharmacists) to evaluate their perceptions and acceptability regarding COVID-19 vaccinations. The authors reported an overall high rate of acceptability for COVID-19 vaccinations of 78.5%, with older age, confidence in vaccine safety, and perceived reliability of vaccine information from the Greek public health authorities being the predictors of acceptability.

Major comments

  1. How were the participant groups being decided? Why nurses and allied health professionals (for examples physiotherapists) were not included in this survey?
  2. Clarify whether the logistic regression analysis performed was bivariate analysis (unadjusted) or multivariate analysis (adjusted). Provide the details at Section 2.3 Statistical analysis, the relevant Result section, and title for Table 3.
  3. In the Methods section, it mentioned that reasons for COVID-19 vaccination refusal were asked. However, these results have not been presented in the manuscript. This needs to be presented and discussed.
  4. Presentation of Table 2 should be improved:
    1. Add another column for the percentage of response for all questions, instead of including them in selected questions within the first column
    2. It is peculiar to half a person (0.5) for a few questions, such as Was there an option of “Neutral”? If that’s the case, this should be explained in the Methods section and that shouldn’t be counted as half fully agree/agree and fully disagree/disagree.  
    3. Keep the digits to 2 decimal places consistently for age

Minor comments

  1. Line 91: Remove “two-hundred thirteen”, and the digit “213” only
  2. Line 94: Change “… the vaccines are in general are safe ….” to “… the vaccines are generally safe…”
  3. Line 100-101, Remove “the” in front of “mass media” and “social media/blogs”. Remove “the Television, radio and newspapers”, which is repetitive of the contents in the brackets.

Author Response

I would like to the Reviewer for their thoughtful and constructive comments which helped us to considerably improve the overall quality of our manuscript. 

Point by point response letter to reviewer’s comments. Manuscript ID: vaccines-1115251

1st REVIEWER

Major comments

  1. How were the participant groups being decided? Why nurses and allied health professionals (for examples physiotherapists) were not included in this survey?

Response: We would like to thank the reviewer for constructive comments which helped us to improve the overall quality of our manuscript. The aim of the study was to evaluate the COVID-19 vaccination acceptance among physicians, dentists and pharmacists. Especially, for dentists and pharmacist’s relevant data are sparse. Nevertheless, we acknowledge the absence of relevant data for other health care professions (e.g. nurses) and we have included a comment in the discussion of the limitations of our survey.

  1. Clarify whether the logistic regression analysis performed was bivariate analysis (unadjusted) or multivariate analysis (adjusted). Provide the details at Section 2.3 Statistical analysis, the relevant Result section, and title for Table 3.

Response: We would like to thank the reviewer for this valuable comment. The logistic regression analysis was not a bivariate but a binary one in which we have adjusted for several confounding factors. The following paragraph has been included into the revised version of the manuscript (methods; statistical analysis): “Variables found to be statistically significant in the univariate analysis (p < 0.05) were included in a stepwise binary logistic regression analysis model, in order to explore potential independent associations with respondents' intention to accept vaccination against COVID-19. In this model COVID-19 vaccination acceptance was the dependent variable and factors found to be significant (p<0.05) in univariate analysis were the independent variables”. The results section and table label have been modified, accordingly

  1. In the Methods section, it mentioned that reasons for COVID-19 vaccination refusal were asked. However, these results have not been presented in the manuscript. This needs to be presented and discussed.

Response: We have revised the text, accordingly, and we have mentioned these data in the results section of the revised version of the manuscript.

  1. Presentation of Table 2 should be improved:
  • Add another column for the percentage of response for all questions, instead of including them in selected questions within the first column

Response: We have modified the presentation of the table, but we are unable to add a new column, since the table is already too big for the space.

  • It is peculiar to half a person (0.5) for a few questions, such as Was there an option of “Neutral”? If that’s the case, this should be explained in the Methods section and that shouldn’t be counted as half fully agree/agree and fully disagree/disagree.

Response: We have corrected the typing error. We didn’t include an option of “Neutral”.

  • Keep the digits to 2 decimal places consistently for age

 Response: The table has been revised, accordingly.

Minor comments

  1. Line 91: Remove “two-hundred thirteen”, and the digit “213” only

Response: Done

  1. Line 94: Change “… the vaccines are in general are safe ….” to “… the vaccines are generally safe…”

Response: We have modified the text, accordingly

  1. Line 100-101, Remove “the” in front of “mass media” and “social media/blogs”. Remove “the Television, radio and newspapers”, which is repetitive of the contents in the brackets.

       Response: Done.

Reviewer 2 Report

Estimated Authors,

Estimated Editors,

first, thank you for the opportunity to review this interesting paper from Papagiannis et al. dealing with Acceptability of Covid-19 Vaccination among Greek health professionals.

As for January 2021, the content of this paper is significant, and the potential implications as well.

Unfortunately, the overall quality of this paper is, at the moment, inadequate for the eventual publication. More precisely:

1) first and foremost: the overall quality of the English text is insufficient, not only in terms of grammar and phrasing. I warmly suggest the revision of the main text by a professional editor;

2) the statistical analysis should be extensively revised: Authors performed a preliminary univariate analysis, followed by a multivariate one (i.e. binary logistic regression). Unfortunately, the model is not described across the text (see the following article for a similar modelling: https://pubmed.ncbi.nlm.nih.gov/32708662/) as it should be. in other words: have you performed an a priori analysis? or have you included in your analyses only factors that in univariate analysis were associated with your outcome? Please fix it.

3) Some variables were initially coded by means of a Likert scale, then were dichotomized in order be more properly fitting bivariate analysis by means of the chi squared test and binary logistic regression; it is correct (see aforementioned example) but this strategy must be reported in the methods section;

4) the sample size formula should be reformatted in order to be more clearly readable;

5) in general, your report on results lack of significant details and should be reformulated accordingly;

6) when dealing with multivariate analyses, your report is misleading. Even though "age (OR= 0.96; 95% C.I. =0.93-0.99), absence of fear over vaccine safety (OR=4.09; 95% C.I.=1.36-12.3) and information received from the Greek public health authorities (OR=11.14;95% C.I.=5.48-22.6) were independently associated with the likelihood of COVID- 19 vaccination acceptance)", the meaning of such variables is quite different. Increasing age (i.e. by unit year) reduced the overall acceptance of the vaccination while other factors, i.e. absence of fear over vaccine safety and information received from the PH authorities INCREASE the acceptance. Please fix accordingly.

7) Tables should be reformatted across the text, accordingly to the instruction to the Authors;

8) Results are quite short, do not discuss the actual evidences (i.e. explanatory variables that were actually associated with the outcome variable) and deserve a long section to the role of pharmacists, that represented less than 13% of total sample. Please reformulate.

Author Response

I would like to thanks  Reviewer for their thoughtful and constructive comments which helped us to considerably improve the overall quality of our manuscript.

Point by point response letter to reviewer’s comments. Manuscript ID: vaccines-1115251

2nd REVIEWER

  • first and foremost: the overall quality of the English text is insufficient, not only in terms of grammar and phrasing. I warmly suggest the revision of the main text by a professional editor;

Response: We would like to thank the reviewer for constructive comments which helped us to improve the overall quality of our manuscript.  We have tried to improve the quality of the English language.

  • the statistical analysis should be extensively revised: Authors performed a preliminary univariate analysis, followed by a multivariate one (i.e. binary logistic regression). Unfortunately, the model is not described across the text (see the following article for a similar modelling: https://pubmed.ncbi.nlm.nih.gov/32708662/) as it should be. in other words: have you performed an a priori analysis? or have you included in your analyses only factors that in univariate analysis were associated with your outcome? Please fix it.

 Response: We would like to thank the reviewer for the comment. We have modified the text, accordingly. In particular, the following sentence has been added into the statistical analysis subsection of the manuscript: Variables found to be statistically significant in the univariate analysis (p < 0.05) were included in a stepwise binary logistic regression analysis model, in order to explore potential independent associations with respondents' intention to accept vaccination against COVID-19. In this model COVID-19 vaccination acceptance was the dependent variable and factors found to be significant (p<0.05) in univariate analysis were the independent variables. Age was transformed to a categorical variable based on the cut-off level of mean age value.

Last, we have also included the following reference: Riccò M Et al. Knowledge, Attitudes, Practices (KAP) of Italian Occupational Physicians towards Tick Borne Encephalitis Trop Med Infect Dis. 2020 Jul 16;5(3):11.

  • Some variables were initially coded by means of a Likert scale, then were dichotomized in order be more properly fitting bivariate analysis by means of the chi squared test and binary logistic regression; it is correct (see aforementioned example) but this strategy must be reported in the methods section;

 Response: Indeed, the majority of variables were initially coded in a four option Likert scale (Fully agree/agree/ disagree/fully disagree). However, these four response options were collapsed into a dichotomous (binary) variable (Fully agree/agree vs. disagree/fully disagree). The following paragraph was included in the revised section of the manuscript (methods; subsection statistical analysis): “The majority of independent variables were initially coded in a four option Likert scale (Fully agree/agree/ disagree/fully disagree). These four response options were collapsed into a dichotomous (binary) variable (Fully agree/agree vs. disagree/fully disagree).

  • the sample size formula should be reformatted in order to be more clearly readable;

 Response:  Done

  • in general, your report on results lack of significant details and should be reformulated accordingly;

Response: Done.

  • when dealing with multivariate analyses, your report is misleading. Even though "age (OR= 0.96; 95% C.I. =0.93-0.99), absence of fear over vaccine safety (OR=4.09; 95% C.I.=1.36-12.3) and information received from the Greek public health authorities (OR=11.14;95% C.I.=5.48-22.6) were independently associated with the likelihood of COVID- 19 vaccination acceptance)", the meaning of such variables is quite different. Increasing age (i.e. by unit year) reduced the overall acceptance of the vaccination while other factors, i.e. absence of fear over vaccine safety and information received from the PH authorities INCREASE the acceptance. Please fix accordingly.

Response: We would like to thank the reviewer for this comment. Indeed, we acknowledge that the way we presented age in the logistic regression analysis was misleading. According to the results of the univariate analysis (Table 2) the mean age of the participants who reported acceptance of the COVID-19 vaccination was significantly higher than the mean age of the respondents who reported no acceptance of the vaccine. In order to improve the interpretability and simplicity of logistic regression results (Table 3) we converted age to a dichotomous variable based on the mean age (45 years). Health care workers at age group > 45 years recorded a twofold likelihood (OR=2.01; 95% C.I. =1.02-4.3) of accepting vaccination in comparison to their colleagues at age group ≤ 45 years (reference category)

  • Tables should be reformatted across the text, accordingly to the instruction to the Authors;

Response: The tables have been revised, accordingly.

  • Results are quite short, do not discuss the actual evidences (i.e. explanatory variables that were actually associated with the outcome variable) and deserve a long section to the role of pharmacists, that represented less than 13% of total sample. Please reformulate.

 Response: We have expanded more on the discussion of the impact of age, fear of side effects and source of information and we have considerably shortened the text related to the pharmacists.

Round 2

Reviewer 1 Report

All comments have been adequately addressed.

Author Response

We would like to thank again the reviewer for persisting efforts to improve the quality of our manuscript.

Reviewer 2 Report

Estimated Authors,

Estimated Editors,

I've appreciated the considerable efforts paid by the Authors to improve the quality of their paper. However, in my opinion, some further improvements are still required, as follows:

1) the main issue: binary logistic regression (Table 3) include a dependent/outcome variable, that in this case is acceptance of COVID-19 vaccine, and a series of explanatory variables, that contribute to defining the model. As you explained in methods section, BLR should have included all variables that in univariate analysis were associated with the outcome variable having p < 0.05; however it is rather unclear how the items included in table 2 were summarized and fitted to the model. Please, reform the methods section in order to better explain such step.

2) minor shortcomings:

a) the paper is not fully compliant with the instructions to the Authors of MDPI journals; tables should be moved across the text and not placed at the end;

b) some minor typos and inconsistent use of acronyms (mainly HCW; after its inception, it should reported consistently)

c) the overall quality of the English text requires, unfortunately, another step of revision

Finally, a small and humble note: when I recommended "see the following article for a similar modelling: https://pubmed.ncbi.nlm.nih.gov/32708662", I was not recommending the direct citation of the paper (that otherwise deals with a very different infectious disease), but only to see an analysis/reporting way that was very akin to the aims of your study (in other words, you can remove the quotation but please follow the model, see point 1 of this second review)

Author Response

We would like to thank again the reviewer for persisting efforts to improve the quality of our manuscript.

1) the main issue: binary logistic regression (Table 3) include a dependent/outcome variable, that in this case is acceptance of COVID-19 vaccine, and a series of explanatory variables, that contribute to defining the model. As you explained in methods section, BLR should have included all variables that in univariate analysis were associated with the outcome variable having p < 0.05; however, it is rather unclear how the items included in table 2 were summarized and fitted to the model. Please, reform the methods section in order to better explain such step.
Response: We would like to thank the reviewer for persisting efforts to improve the quality of our manuscript. Regarding logistic regression analysis results, we would like to clarify that-in order to save space- we have presented only these results which have reached statistical significance.

2) minor shortcomings:
a) the paper is not fully compliant with the instructions to the Authors of MDPI journals; tables should be moved across the text and not placed at the end;
Response: Done

b) some minor typos and inconsistent use of acronyms (mainly HCW; after its inception, it should reported consistently)
Response: Done

c) the overall quality of the English text requires, unfortunately, another step of revision
Response: We have tried to improve the quality of the English language and we would like to inform you that at least two of the authors [(GR) and (FM)] are native English speakers.
Finally, a small and humble note: when I recommended "see the following article for a similar modelling: https://pubmed.ncbi.nlm.nih.gov/32708662", I was not recommending the direct citation of the paper (that otherwise deals with a very different infectious disease), but only to see an analysis/reporting way that was very akin to the aims of your study (in other words, you can remove the quotation but please follow the model, see point 1 of this second review)
Response: Thank you for your comment
